# Skeletal muscle-kidney crosstalk in a cohort of critical illness survivors

Heitor S. Ribeiro[ID][1]*, Dário R. Mondini[ID][1], Guilherme P. Santa-Catharina[1], Lia Marçal[2], Leila Antonângelo[2], Luis Yu[1], Dirce M. T. Zanetta[3], Linamara R. Battistella[ID][4], Geraldo F. Busatto[5], Carlos R. R. Carvalho[ID][6], Emmanuel A. Burdmann[1], for the HCFMUSP COVID-19 Study Group[¶]

1 Laboratório de Investigação (LIM) 12, Serviço de Nefrologia, Faculdade de Medicina da Universidade de São Paulo, São Paulo, Brazil, 2 LIM 03, Faculdade de Medicina, Universidade de São Paulo, Sao Paulo, Brazil, 3 Departamento de Epidemiologia, Faculdade de Saude Publica, Universidade de Sao Paulo, Sao Paulo, Brazil, 4 Departamento de Medicina Legal, Bioética, Medicina do Trabalho e Medicina Física e Reabilitação, Hospital das Clinicas HCFMUSP, Faculdade de Medicina, Universidade de Sao Paulo, Sao Paulo, Brazil, 5 Departamento e Instituto de Psiquiatria, Hospital das Clinicas HCFMUSP, Faculdade de Medicina, Universidade de São Paulo, Sao Paulo, Brazil, 6 Divisão de Pneumologia, Instituto do Coração, Hospital das Clinicas HCFMUSP, Faculdade de Medicina, Universidade de Sao Paulo, Sao Paulo, Brazil

¶ Membership of the HCFMUSP COVID-19 Study Group is listed in the Acknowledgments.
* heitorribeiro@usp.br

## Abstract

### Background

The skeletal muscles and kidneys are frequently affected during critical illness; however, their crosstalk remains poorly explored, especially in the long-term evolution. Therefore, we investigated the crosstalk between skeletal muscle and kidney function in COVID-19 survivors.

### Methods

A cross-sectional analysis of a prospective cohort study with survivors of moderate to severe COVID-19 hospitalization. Skeletal muscle assessments included hand-grip strength, calf circumference, ultrasound-measured quadriceps thickness, and gait speed test. Sarcopenia was diagnosed by modified EWGSOP2 (low handgrip strength *plus* low ultrasound-measured quadriceps thickness). Kidney function was assessed by estimated glomerular filtration rate (eGFR), urinary albumin-to-creatinine ratio (UACR), and urine sediment analysis. Abnormal kidney function was defined as an eGFR < 60 mL/min/1.73m$^2$, albuminuria (≥30 mg/g), and/or leukocyturia or hematuria.

### Results

A total of 734 survivors (46% female, 43% ≥60 years, 35% with diabetes) were assessed 7 ± 2 months post-hospital discharge. Sarcopenia was diagnosed in 21.4%

**Data availability statement:** Data used in this manuscript is part of a large cohort, HCFMUSP COVID-19 Study Group, and is available as Supporting Information.

**Funding:** This study receives funding from the Fundação de Amparo à Pesquisa do Estado de São Paulo (FAPESP; grant 22/01769-5) and from Instituto Todos pela Saúde (ITpS; grant C1721). EAB receives a research grant (Bolsa de Produtividade em Pesquisa, 304743/2017-8) from The National Council for Scientific and Technological Development (CNPq). HSR received a postdoctoral scholarship from FAPESP (24/04564-0).

**Competing interests:** EAB received speaker fees from AstraZeneca, Baxter, and Fresenius outside of the submitted work. Other authors declare that they have no competing interests.

of the cohort. Positive significant associations with eGFR were observed for calf circumference ($\beta = 0.42$ ml/min/1.73m$^2$, 95%CI: 0.06 to 0.78) and thicknesses (mm) of rectus femoris ($\beta = 0.47$, 95%CI: 0.01 to 0.94) and vastus intermedius ($\beta = 0.55$, 95%CI: 0.14 to 0.96). None of the skeletal muscle parameters were associated with UACR (mg/g). Survivors with sarcopenia had lower eGFR (–5.8 ml/min/1.73m$^2$, 95%CI: –10.8 to –0.9), but similar frequencies of low eGFR (24% *vs*. 18%; $p = 0.137$), albuminuria (27% *vs*. 31%; $p = 0.434$) and abnormal urine sediment (17% *vs*. 22%; $p = 0.217$) as compared to those without sarcopenia. Sarcopenia was not associated with higher odds of low eGFR, albuminuria, or abnormal urine sediment.

## Conclusions

In survivors of moderate to severe COVID-19 hospitalization, skeletal muscle mass was associated with eGFR, whereas sarcopenia *per se* was not independently associated with poor kidney function. These findings suggest the existence of a skeletal muscle-kidney crosstalk in this population.

## Introduction

The long-term effects of SARS-CoV-2 infection, commonly referred to as long coronavirus disease (COVID), have gathered attention in the care of COVID-19 survivors [1,2]. Multiple organs are affected in the long COVID, including the skeletal muscles and kidneys [3]. A systematic review showed that 25% of COVID-19 survivors presented muscle weakness as a post-acute sequela [4]. Similarly, some studies with COVID-19 survivors have shown a persistent long-term alteration of kidney function [5,6].

The kidneys are receptors of signaling molecules of the skeletal muscle, which may affect the metabolism of proteins [7]. Moreover, myokines derived from the skeletal muscle have been shown to attenuate kidney damage and fibrosis [8]. Serum creatinine (sCr), the most used biomarker for estimated glomerular filtration rate (eGFR), reflects both skeletal muscle mass and kidney function. Previous studies with older adults showed an association of eGFR and albuminuria with muscle strength and muscle mass [9–11], which are both components of sarcopenia, one of the most well-recognized skeletal muscle diseases [12].

Sarcopenia is strongly associated with adverse kidney outcomes, such as rapid eGFR decline, and incidence and progression of chronic kidney disease (CKD) and albuminuria [13–15]. Therefore, post-acute skeletal muscle alterations in the long COVID [16], including sarcopenia, might negatively impact long-term kidney function.

It is unknown if this association between skeletal muscle and kidney function occurs in survivors of COVID-19, and the muscle-kidney crosstalk has yet to be explored in this population. To address this knowledge gap, we investigated the association between skeletal muscle parameters and kidney function markers in a prospective cohort of moderate to severe survivors of COVID-19 hospitalization.

We hypothesize that skeletal muscle parameters are associated with kidney function markers, and that COVID-19 survivors with sarcopenia will have worse eGFR, and higher frequencies of albuminuria and abnormal urine sediment.

## Materials and methods

### Study design, setting, and participants

This is a cross-sectional analysis of the follow-up in-person visit of the *Hospital das Clínicas, Faculdade de Medicina, Universidade de São Paulo* (HCFMUSP) COVID-19 prospective cohort study. The study protocol with detailed information was published elsewhere [17]. In brief, survivors of hospitalization during the first COVID wave (between March and August 2020) from the largest academic hospital in Brazil were invited for a follow-up in-person visit between October 2020 and April 2021. Eligibility criteria were laboratory-confirmed COVID-19 diagnosis, hospital stay longer than 24 hours, and 18 years or older. Exclusion criteria included previous diagnosis of dementia or end-stage cancer, living in long-term care facilities, insufficient physical mobility to leave home during follow-up assessments, kidney failure with replacement therapy, and serum creatinine (sCr) or handgrip strength not assessed at the follow-up visit. Research protocols were approved by the local Ethics Committees of the HCFMUSP (#4.270.242, #4.502.334, #4.524.031, #4.302.745, and #4.391.560) and patients provided written informed consent. Data were accessed for research purposes in 01/07/2023. This manuscript used data from the cross-sectional follow-up in-person visit and was reported following the Strengthening the Reporting of Observational Studies in Epidemiology Statement [18].

### Data collection

Patients were evaluated for up to 11 months after hospital discharge. In summary, a teleconsulting semi-structured interview (collection of sociodemographic characteristics, lifestyle habits, and medical history), and an in-person visit (objective physical assessments and laboratory test) were performed. Data were stored in the Research Electronic Data Capture (REDCap) system.

The collected cross-sectional data included older age (≥ 65 years), ethnicity, body mass index classified according to the World Health Organization [19], comorbidities (kidney disease included other than CKD), kidney function, and skeletal muscle parameters.

### Skeletal muscle assessment

Skeletal muscle assessments included handgrip strength, calf circumference, ultrasound-measured quadriceps thickness, and gait speed test. Assessments were conducted by an expert research team under the supervision of L.R.B and detailed elsewhere [20].

**Handgrip strength.** Handgrip strength was measured with a Jamar® hydraulic hand dynamometer (Sammons Preston, Bolingbrook, Illinois, USA) with individuals seated, shoulder in a neutral position with elbows flexed at 90∘ to the body position and neutral wrist position. Three measurements were performed for both sides, and the mean score of the side with the highest score was recorded in kg. Low handgrip strength was defined as <27 kg for males and <16 kg for females [21].

**Calf circumference.** As a marker of muscle mass, calf circumference was measured using a non-stretchable measuring tape.

**Quadriceps muscle thickness.** A portable ultrasound system evaluated the thickness of the right rectus femoris and vastus intermedius muscles in millimeters (mm). Muscle thickness was measured by the perpendicular distance between the superior and inferior muscle fascia, at the largest diameter without the surrounding fascia, subcutaneous tissue, and skin. Detailed protocol has been previously published [22]. To diagnose low skeletal muscle mass, given the absence of standardized cut-off values for quadriceps muscle thickness in Brazil, we adopted sex-specific lowest tertiles (p33) as

follows: (i) rectus femoris <11.6 mm for males and <9.8 mm for females; and (ii) vastus intermedius <11.4 mm for males and <10.3 mm for females.

**Gait speed test.** Gait speed test was assessed by the usual gait speed in 10 meters. Performed distance divided by time (*i.e.*, speed [m/s]) was considered for analysis [23].

**Sarcopenia diagnosis.** The revised European Working Group on Sarcopenia in Older People (EWGSOP2) was adopted to diagnose sarcopenia in the presence of low muscle strength and low skeletal muscle mass [24]. All patients analyzed had available handgrip strength and ultrasound-measured quadriceps thickness.

## Kidney function assessment

Blood serum and urine samples were collected in the follow-up in-person visit. A urine sample was used for assessment of urine sediment and a blood sample was used for sCr evaluation, conducted by the HCFMUSP laboratory. A urine sample was stored in a freezer at −80ºC until assayed by a specialized laboratory at the university by L.M and L.A (urinary creatinine and albumin). Abnormal kidney function was diagnosed according to the Kidney Disease Improving Global Outcomes (KDIGO) guideline [25].

**Low eGFR.** Estimated glomerular filtration rate (eGFR) was calculated using the CKD Epidemiology Collaboration (CKD-EPI 2021) race-free sCr-based equation [26]. Low eGFR was defined as < 60 mL/min/1.73 m$^2$ [25].

**Albuminuria.** Albuminuria was considered as a urinary albumin-to-creatinine ratio (UACR) ≥ 30 mg/g [25].

**Abnormal urine sediment.** Abnormal urine sediment was defined as hematuria (> 3 red blood cells/field) and/or leukocyturia (> 10 white blood cells/field) [25].

## Statistical analyses

**Sample size.** Sample size calculations were not performed because we recruited as many survivors as possible.

**Missing data and imputation.** Missing data is detailed in Supplementary S2 Table, and imputation was not performed. Frequency variables are described as valid percentages.

**Descriptive analysis.** Continuous data are shown as mean and standard deviation or median and interquartile range (IQR), depending on data normality. Normality of data was assessed by histogram visual inspection and the Shapiro-Wilk test. Comparisons between the groups with and without sarcopenia were conducted using unpaired the Student's t-test or the Mann-Whitney U test, depending on data normality. For categorical variables, we performed the Chi-Square or Fisher Exact tests.

**Association measures.** Associations between independent (handgrip strength, calf circumference, gait speed, and quadriceps muscle thickness) and dependent variables (eGFR and UACR) were tested by running linear regression analysis and reported as β (beta) and 95% confidence interval (CI). The model was adjusted for clinically relevant confounders (age, sex, and diabetes). Sensitivity analyses were conducted by excluding younger (< 60 years) and male survivors.

Binary logistic regressions were conducted to investigate the association between sarcopenia (probable + confirmed + severe stages; reference = no sarcopenia) and individual components of abnormal kidney function (low eGFR, albuminuria, and abnormal urine sediment). Odds ratios (OR) and 95% CI were calculated. Adjusted model included clinically relevant confounders; older age (reference = <60 years), sex (reference = male), diabetes (reference = no diabetes), and ICU admission during hospitalization (reference = no admission).

Analyses were performed using the Statistical Package for the Social Sciences (version 29.0, SPSS Inc, Chicago, USA) and GraphPad Prism (version 8.4, GraphPad Software, San Diego, USA). A two-tailed *p*-value < 0.05 was considered statistically significant.

## Results

### Recruitment

A total of 870 survivors were included in the HCFMUSP COVID-19 cohort. After applying exclusion criteria, 734 were considered for this cross-sectional study (see Supplementary S1 Fig for the flowchart). Survivors were assessed 211 [IQR: 172–258] days post-hospital discharge. The mean age of the analyzed survivors was 55.2 ± 14.0 years, 43.2% were ≥ 60 years, 46.0% were female, and 56.8% had obesity.

### Skeletal muscle health

Sarcopenia was diagnosed in 21.4% (n = 157) of the survivors. Table 1 shows that patients with sarcopenia were more frequently aged ≥60 years (50% vs. 41%; p = 0.046) and were more vaccinated to COVID (23% *vs.* 13%; p = 0.035) compared to those without sarcopenia.

### Kidney function

At hospital admission and discharge, eGFR values were similar between groups (p-values = 0.235 and 0.252, respectively). Markers of kidney function are shown in Table 2. Survivors with sarcopenia had lower eGFR (–5.8 ml/min/1.73m$^2$, 95%CI: –10.8 to –0.9), but similar UACR (p = 0.653) values compared to those without sarcopenia. Regarding abnormal kidney function, a low eGFR (< 60 mL/min/1.73 m$^2$) was present in 19.2% (n = 141) of patients, whereas 30.3% (n = 177) showed albuminuria (149 missing data), and 20.8% (n = 147) had an abnormal urine sediment (26 missing data; 14.3% leukocyturia and 9.5% hematuria). Fig 1 shows that survivors with sarcopenia had similar prevalence of low eGFR (p = 0.137), albuminuria (p = 0.434), and abnormal urine sediment (p = 0.217) compared to those without sarcopenia.

### Association between skeletal muscle parameters and kidney function markers

Linear regression analysis adjusted for age, sex, and diabetes is described in Table 3. Positive and significant associations with eGFR were only found for calf circumference (β = 0.42 ml/min/1.73m$^2$, 95%CI: 0.06 to 0.78), and thicknesses (mm) of rectus femoris (β = 0.47, 95%CI: 0.01 to 0.94) and vastus intermedius (β = 0.55, 95%CI: 0.14 to 0.96, respectively). In sensitivity analyses excluding younger and male survivors (Supplementary S2 Table), no associations remained significant. In the overall analysis (Table 3), none of the skeletal muscle parameters were associated with UACR (mg/g). However, gait speed (sec) was associated with UACR in females (β = 1.33 mg/g, 95%CI: 0.14 to 2.52) in a sensitivity analysis (Supplementary S3 Table).

Binary logistic regression (Fig 2) adjusted for older age, female sex, diabetes, and ICU admission during hospitalization showed that sarcopenia was not associated with low eGFR (OR = 1.31, 95% CI: 0.84 to 2.03), albuminuria (OR = 0.85, 95% CI: 0.54 to 1.34), or abnormal urine sediment (OR = 0.75, 95% CI: 0.47 to 1.21).

## Discussion

### Main findings

In this cross-sectional study, we investigated the association between late skeletal muscle and kidney function in survivors of moderate to severe COVID-19 hospitalization. Quadriceps muscle thickness and calf circumference – parameters of skeletal muscle mass – were associated with eGFR, while physical function was not. Sarcopenia was found in one out of five survivors. Survivors with sarcopenia had lower eGFR, but showed similar rates of low eGFR, albuminuria, and abnormal urine sediments. Overall, our findings suggest an existing skeletal muscle-kidney cross-talk in survivors of COVID-19 critical illness, although the association is heterogeneous and primarily manifested through lower eGFR.

**Table 1. Characteristics according to sarcopenia status.**

| | All patients (n = 734) | Sarcopenia (n = 157) | No sarcopenia (n = 577) | p-value |
|---|---|---|---|---|
| Age (years) | 55.2 ± 14.0 | 57.5 ± 14.3 | 54.6 ± 13.9 | 0.022 |
| Older age (≥ 60 years), n (%) | 317 (43.2) | 79 (50.3) | 238 (41.2) | 0.046 |
| Female, n (%) | 338 (46.0) | 65 (41.4) | 273 (47.3) | 0.206 |
| **Ethnicity**, n (%) | | | | 0.502 |
| White | 344 (47.2) | 68 (43.6) | 276 (48.2) | |
| Black | 100 (13.7) | 24 (15.4) | 76 (13.3) | |
| Mixed | 268 (36.8) | 62 (39.7) | 206 (36.0) | |
| Indigenous | 7 (1.0) | 0 (0) | 7 (1.2) | |
| Not reported | 10 (1.4) | 2 (1.3) | 8 (1.4) | |
| **Body mass index**, n (%) | | | | 0.706 |
| Underweight | 6 (0.9) | 2 (1.4) | 4 (0.7) | |
| Normal weight | 79 (11.6) | 14 (9.7) | 65 (12.1) | |
| Overweight | 209 (30.7) | 42 (29.2) | 167 (31.1) | |
| Obese | 387 (56.8) | 86 (59.7) | 301 (56.1) | |
| **Comorbidities**, n (%) | | | | |
| Hypertension | 408 (56.7) | 85 (55.2) | 323 (57.1) | 0.714 |
| Diabetes | 251 (34.9) | 51 (33.1) | 200 (35.3) | 0.635 |
| Respiratory disease | 54 (7.5) | 16 (10.4) | 38 (6.7) | 0.165 |
| Kidney disease | 50 (7.0) | 4 (2.6) | 46 (8.1) | 0.018 |
| Cancer | 27 (4.1) | 7 (4.9) | 20 (3.8) | 0.632 |
| eGFR at hospital admission (mL/min/1.73 m$^2$) | 76.9 ± 34.8 | 73.9 ± 33.0 | 77.7 ± 35.2 | 0.235 |
| **Outcomes during hospitalization**, n (%) | | | | |
| Hospital length of stay, median [IQR] | 13 [7–23] | 13 [7–24] | 13 [7–23] | 0.998 |
| ICU admission | 424 (58.8) | 95 (61.7) | 329 (58.0) | 0.460 |
| Intubation | 290 (40.6) | 68 (44.4) | 222 (39.5) | 0.307 |
| Acute kidney injury | 576 (78.5) | 125 (79.6) | 451 (78.2) | 0.743 |
| Use of KRT | 93 (12.7) | 17 (10.8) | 76 (13.2) | 0.500 |
| eGFR at hospital discharge (mL/min/1.73 m$^2$) | 84.4 ± 31.9 | 87.1 ± 30.2 | 83.7 ± 32.3 | 0.252 |
| **Events between discharge and follow-up**, n (%) | | | | |
| COVID vaccination | 51/344 (14.8) | 15/64 (23.4) | 36/280 (12.9) | 0.035 |
| Medical appointment | 170 (23.2) | 34 (21.7) | 136 (23.6) | 0.885 |
| Hospital readmission | 63 (8.6) | 13 (8.3) | 50 (8.7) | |

eGFR, estimated glomerular filtration rate; IQR, interquartile range; ICU, intensive care unit; KRT, kidney replacement therapy. There was high missing data for COVID vaccination status because vaccine was not fully available at the time.

## The skeletal muscle-kidney crosstalk

Recent studies have investigated the potential muscle-kidney crosstalk, revealing conflicting results depending on the studied population and the markers used to assess skeletal muscle and kidney function [9–11]. In survivors of critical illness, such as COVID-19, little is known and there is currently no evidence in the long-term. In the general population, the available evidence mostly investigates the association between sarcopenia and abnormal kidney function. In people with diabetes, a systematic review with six studies showed that sarcopenia was significantly associated with albuminuria, low eGFR, and proteinuria [27]. Data from 594 community-dwelling Japanese adults aged ≥40 years demonstrated that lower

**Table 2. Kidney function markers according to sarcopenia status.**

| Kidney function markers | All patients (n = 734) | Sarcopenia (n = 157) | No sarcopenia (n = 577) | p-value |
|---|---|---|---|---|
| eGFR (mL/min/1.73 m²) | 82.2±28.2 | 77.6±27.7 | 83.5±28.2 | 0.021 |
| ≥ 90 | 326 (44.4) | 56 (35.7) | 270 (46.8) | 0.110 |
| 60–89 | 267 (36.4) | 64 (40.8) | 203 (35.2) | |
| 30–59 | 103 (14.0) | 29 (18.5) | 74 (12.8) | |
| 15–29 | 12 (1.6) | 3 (1.9) | 9 (1.6) | |
| < 15 | 26 (3.5) | 5 (3.2) | 21 (3.6) | |
| UACR (mg/g)* | 10.7 [2.9–43.4] | 11.2 [2.4–31.9] | 10.7 [3.5–45.7] | 0.653 |
| < 30 | 408 (69.7) | 86 (72.9) | 322 (69.0) | 0.549 |
| 30–300 | 176 (30.1) | 32 (27.1) | 144 (30.8) | |
| > 300 | 1 (0.2) | 0 (0.0) | 1 (0.2) | |

eGFR, estimated glomerular filtration rate; UACR, urinary albumin-to-creatinine ratio. Values are described in median and interquartile range. * n = 591 due to random missing data.

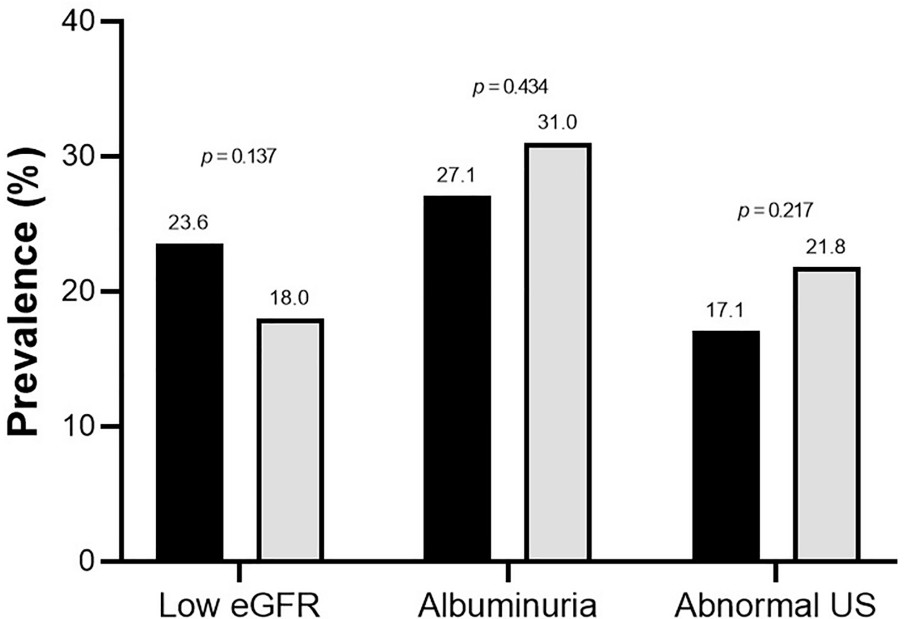

**Fig 1. Abnormal kidney function according to sarcopenia status.** eGFR, estimated glomerular filtration rate; US, urine sediment. Low eGFR was defined as < 60 mL/min/1.73 m²; albuminuria as a urinary albumin-to-creatinine ratio ≥ 30 mg/g; and abnormal urine sediment as leukocyturia and/or hematuria. There was missing data for albuminuria (n = 147) and urine sediment (n = 26).

eGFR was associated with low handgrip strength when using serum cystatin C, but not sCr [9]. This suggests that different markers of kidney function may influence the observed muscle-kidney crosstalk. For instance, sCr originates from skeletal muscle metabolism, and eGFR equations based on sCr are significantly influenced by variations in skeletal muscle mass [28]. The reduction in muscle mass frequently observed in critical illnesses, like COVID-19, may compromise the reliability of sCr as a primary kidney function marker [29]. However, in our cohort, survivors were assessed 7±2 months after hospital discharge, a period when sCr levels are expected to have reached a steady state, making it a more reliable marker of kidney function than during the acute phase of the critical illness.

**Table 3. Association between skeletal muscle parameters and kidney function markers.**

| Skeletal muscle | Kidney function | | | |
| --- | --- | --- | --- | --- |
| | eGFR (mL/min/1.73m²) | | UACR (mg/g) | |
| | β (95% CI) | *p*-value | β (95% CI) | *p*-value |
| Handgrip strength (kg) | 0.02 (-0.01 to 0.03) | 0.159 | 0.02 (-0.18 to 0.06) | 0.280 |
| Calf circumference (cm) | 0.42 (0.06 to 0.78) | 0.021 | -0.19 (-0.92 to 0.54) | 0.613 |
| Gait speed (sec) | 0.21 (-0.17 to 0.60) | 0.277 | 0.47 (-0.27 to 1.21) | 0.211 |
| Rectus femoris (mm) | 0.47 (0.01 to 0.94) | 0.049 | 0.15 (-0.77 to 1.08) | 0.744 |
| Vastus intermedius (mm) | 0.55 (0.14 to 0.96) | 0.009 | 0.53 (-0.26 to 1.33) | 0.188 |

CI, confidence interval; eGFR, estimated glomerular filtration rate; UACR, urinary albumin-to-creatinine ratio. Linear regression analysis adjusted for age (continuous), sex, and diabetes.

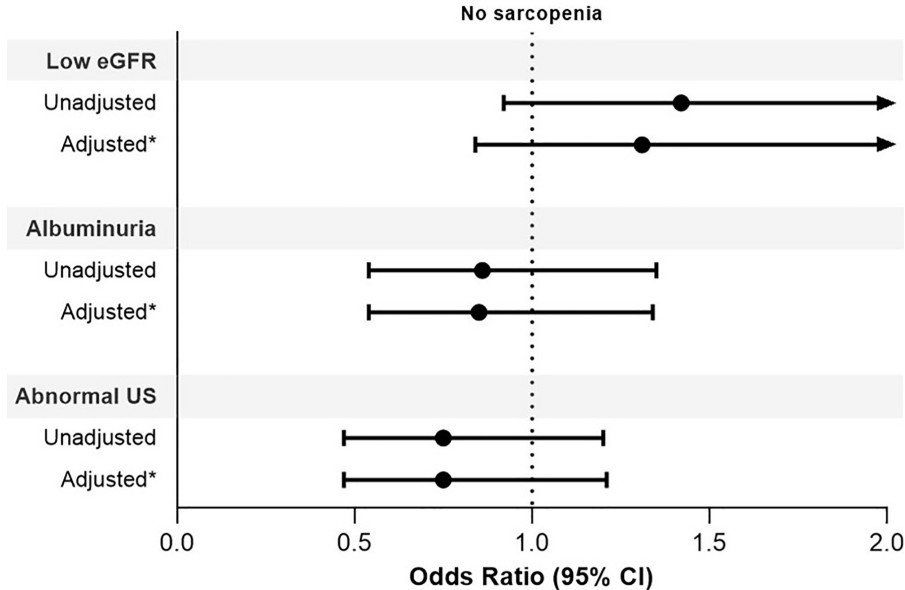

**Fig 2. Association between sarcopenia and abnormal kidney function.** eGFR, estimated glomerular filtration rate; US, urine sediment. * Adjusted model included older age (reference = <60 years), sex (reference = male), diabetes (reference = no diabetes), and intensive care unit admission during hospitalization (reference = no admission). Low eGFR was defined as < 60 mL/min/1.73 m²; albuminuria as a urinary albumin-to-creatinine ratio ≥ 30 mg/g; and abnormal urine sediment as leukocyturia and/or hematuria. There was missing data for albuminuria (n = 147) and urine sediment (n = 26).

Unlike previous studies [10,30,31], we did not observe a significant association between sarcopenia and albuminuria, a well-established marker of glomerular dysfunction and CKD progression [25]. In the study by Kim et al. with 2158 participants of the Korea National Health and Nutrition Examination Survey aged over 19 years [31], those with sarcopenia had a higher prevalence of albuminuria and were three times more likely to exhibit it. In this study, sarcopenia was defined by two standard deviation of skeletal muscle index below a sex-specific mean for a younger reference group, different from our definition mainly based on low handgrip strength, which is aligned with recent sarcopenia guidelines [32]. Hara et al., in a population-based study with 916 Japanese participants aged ≥40 years [10], found an independent negative association between handgrip strength (kg) and log-transformed UACR in male, female, and older participants; with no significant association in youngers. The authors, however, did not investigate the association of sarcopenia (*i.e.*, low muscle strength and/or low muscle mass) with albuminuria. Although we did not find such a significant association in our study

with survivors of COVID-19 critical illness, evidence from other populations suggests this interaction. The variability in findings highlights the inconclusive nature of this topic, warranting further investigation. The mechanisms that may explain the association between skeletal muscle health and albuminuria are the common shared pathophysiological processes of insulin resistance, low-grade chronic inflammation, and oxidative stress.

## Clinical applicability

Muscle weakness and fatigue are common post-acute sequelae of COVID-19 [4]. Our findings, highlighting the significant skeletal muscle-kidney crosstalk in survivors of moderate to severe COVID-19 hospitalization, could assist clinicians in designing targeted interventions to address long-term kidney function decline and mitigate CKD progression. Exercise and nutritional interventions are well-documented strategies to maintain and/or improve skeletal muscle outcomes, as well as recommended by the KDIGO as important means for delaying CKD progression and managing its complication [25]. However, whether improvements in skeletal muscle health may positively affect kidney function in survivors of critical illness has yet to be explored, which we hope to investigate in upcoming research projects within this cohort of moderate to severe COVID-19 survivors.

## Strengths and limitations

The strengths of this study rely on the large sample size in a real-world setting. In addition, unlike other studies on long COVID, outcomes of interest were objectively determined, while previous studies mainly conducted phone interviews with self-reported measures of physical function. Previous studies had only estimated skeletal muscle mass, while we measured quadriceps thickness and explored its association with various kidney function markers.

Despite these strengths, limitations exist. The study was conducted at a single public academic health center, and the findings might lack generalizability. The exclusion of survivors with insufficient physical mobility to leave home may have underestimated cases of sarcopenia. eGFR was estimated by using sCr, a biomarker derived from skeletal muscle metabolism, thus other markers such as serum cystatin C could have resulted in different patterns of muscle-kidney crosstalk. By assuming this bias, it likely overestimates kidney function in survivors with sarcopenia, thus underestimating CKD prevalence and severity. Pre-hospitalization markers of skeletal muscle and kidney function were unavailable, potentially introducing bias from undiagnosed cases of sarcopenia and/or abnormal kidney function before COVID-19 hospitalization. In order to minimize this bias, patients with kidney failure undergoing replacement therapy were excluded. Our cohort consists of survivors of moderate to severe COVID-19, and generalization to other critical illness models should be done with caution, as COVID-19 primarily manifests pulmonary complications.

## Conclusions

In survivors of moderate to severe COVID-19 hospitalization, skeletal muscle mass was associated with eGFR, whereas sarcopenia *per se* was not independently associated with poor kidney function. These findings suggest an interaction between skeletal muscle and kidney function.

Future longitudinal prospective research is needed to establish a causal relationship between sarcopenia and incident CKD in survivors of critical illnesses, which is currently being conducted by our group in this cohort of COVID-19 survivors.

## Supporting information

**S1 Fig. Flowchart.**
(DOCX)

**S2 Table. Missing data.** eGFR. estimated glomerular filtration rate.
(DOCX)

**S3 Table. Sensitivity analysis for the association between skeletal muscle parameters and kidney function markers.** CI, confidence interval; eGFR, estimated glomerular filtration rate; UACR, urinary albumin-to-creatinine ratio. Multiple linear regression analysis adjusted for age (continuous), sex, and diabetes. We defined younger survivors as those <60 years. Variables considered for the sensitivity analyses were not included in the adjusted model.
(DOCX)

**S4 Checklist. STROBE Statement—Checklist of items that should be included in reports of cohort studies.**
(DOCX)

**S5 Database. Database of the study.**
(XLSX)

## Acknowledgments

We thank Laura S. Azevedo for her wide support in data management, Caroline S. Faria for helping in the urinary bio-markers analysis, and Fábio Augusto R. Gonçalves for helping in the initial database analysis.

During the preparation of this work, the authors used ChatGPT in order to rephrase content previously published by our study group on the same project. After using this tool, the authors reviewed and edited the content as needed and take full responsibility for the content of the publication.

HCFMUSP COVID-19 Study Investigators: Marta Imamura, PhD[4]; Caroline S. Faria, PhD[2]; Laura S. Azevedo[6]; Fábio Augusto R. Gonçalves[1]. Carlos R. R. Carvalho is the lead author for this group (carlos.carvalho@hc.fm.usp.br).

## Author contributions

**Conceptualization:** Heitor S. Ribeiro, Dário R. Mondini, Luis Yu, Geraldo F. Busatto, Carlos R. R. Carvalho, Emmanuel A. Burdmann.

**Data curation:** Heitor S. Ribeiro, Guilherme P. Santa-Catharina, Lia Marçal, Leila Antonângelo, Linamara R. Battistella.

**Formal analysis:** Heitor S. Ribeiro, Leila Antonângelo, Dirce M. T. Zanetta.

**Funding acquisition:** Geraldo F. Busatto, Carlos R. R. Carvalho, Emmanuel A. Burdmann.

**Investigation:** Heitor S. Ribeiro, Leila Antonângelo, Luis Yu, Linamara R. Battistella, Geraldo F. Busatto, Carlos R. R. Carvalho, Emmanuel A. Burdmann.

**Methodology:** Heitor S. Ribeiro, Guilherme P. Santa-Catharina, Leila Antonângelo, Luis Yu, Dirce M. T. Zanetta, Linamara R. Battistella, Geraldo F. Busatto, Carlos R. R. Carvalho, Emmanuel A. Burdmann.

**Project administration:** Luis Yu, Geraldo F. Busatto, Carlos R. R. Carvalho, Emmanuel A. Burdmann.

**Resources:** Lia Marçal, Geraldo F. Busatto, Carlos R. R. Carvalho, Emmanuel A. Burdmann.

**Software:** Geraldo F. Busatto, Carlos R. R. Carvalho, Emmanuel A. Burdmann.

**Supervision:** Dirce M. T. Zanetta, Carlos R. R. Carvalho, Emmanuel A. Burdmann.

**Validation:** Dirce M. T. Zanetta, Emmanuel A. Burdmann.

**Visualization:** Dirce M. T. Zanetta, Emmanuel A. Burdmann.

**Writing – original draft:** Heitor S. Ribeiro.

**Writing – review & editing:** Dário R. Mondini, Guilherme P. Santa-Catharina, Lia Marçal, Leila Antonângelo, Luis Yu, Dirce M. T. Zanetta, Linamara R. Battistella, Geraldo F. Busatto, Carlos R. R. Carvalho, Emmanuel A. Burdmann.

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
