## [Decision Letter · Decision Letter 0]

10 Sep 2025

Dear Dr. Ribeiro,

Thank you for submitting your manuscript to PLOS ONE. After careful consideration, we feel that it has merit but does not fully meet PLOS ONE’s publication criteria as it currently stands. Therefore, we invite you to submit a revised version of the manuscript that addresses the points raised during the review process.

**ACADEMIC EDITOR: **

Dear Dr. Ribeiro,

as you will read below, I agree with Reviewer 1 on two major concerns that I believe should be addressed to strengthen the manuscript. First, quadriceps ultrasound, considered a reliable method for assessing muscle mass, was available but not used to define confirmed sarcopenia, whereas calf circumference has known limitations. Second, creatinine-based eGFR likely overestimates kidney function in sarcopenic patients, which may lead to an underestimation of CKD prevalence and severity. This issue should be discussed more explicitly.

We look forward to receiving your revised manuscript.

Kind regards,

Diego Moriconi

Academic Editor

PLOS ONE

Journal Requirements:

“This study receives funding from the Fundação de Amparo à Pesquisa do Estado de São Paulo (FAPESP; grant 22/01769-5) and from Instituto Todos pela Saúde (ITpS; grant C1721). EAB receives a research grant (Bolsa de Produtividade em Pesquisa, 304743/2017-8) from The National Council for Scientific and Technological Development (CNPq). HSR receives a postdoctoral scholarship from FAPESP (24/04564-0).”

“EAB received speaker fees from AstraZeneca, Baxter, and Fresenius outside of the submitted work. Other authors declare that they have no competing interests.”

5. In the online submission form, you indicated that [Data is available at reasonable requests to the corresponding author.].

6. One of the noted authors is a group or consortium [HCFMUSP COVID-19 Study Group]. In addition to naming the author group, please list the individual authors and affiliations within this group in the acknowledgments section of your manuscript. Please also indicate clearly a lead author for this group along with a contact email address.

Additional Editor Comments:

Reviewer #1:

The study by Ribeiro et al. is a single-center cross-sectional study that aimed to assess the relationship between muscle mass and function parameters and kidney health in 743 survivors of the first wave of COVID-19.

While there are published data on muscle-kidney cross-talk, there are no data on this topic in the context of COVID-19, which makes this study original.

My questions and comments are as follows:

- The revised European consensus on the definition and diagnosis of sarcopenia (ref 21) states that ”Although anthropometry is sometimes used to reflect nutritional status in older adults, it is not a good measure of muscle mass [78]. Calf circumference has been shown to predict performance and survival in older people (cut-off point <31 cm) [79]. As such, calf circumference measures may be used as a diagnostic proxy for older adults in settings where no other muscle mass diagnostic methods are available.” There is therefore some uncertainty about the reliability of this method for measuring muscle mass. Since the patients in the study had their quadriceps muscle thickness measured by ultrasound (a method considered reliable for measuring muscle mass), why was the definition of confirmed sarcopenia not based on this ultrasound method?

- The decrease in muscle strength and function may also be due to a change in muscle quality, in addition to muscle quantity (see, for example, 10.1093/ndt/gfy139). Were muscle quality parameters also measured during the quadriceps ultrasound?

- None of the patients in the study had severe sarcopenia. Could this be explained by the following exclusion criterion? “insufficient physical mobility to leave home during follow-up assessments”

- In the results section, on page 10, regarding the following sentence “Table 1 shows that patients with sarcopenia were older (52% vs. 40%; p = 0.002)”, it should be explicitly specified that this refers to patients over the age of 60.

- In the results section, on page 11, regarding the following sentence “Hospital admission and clinical characteristics based on sarcopenia status are described in Supplementary Table S3”, It would be useful to specify in the manuscript that at the time of hospitalization for COVID-19 infection, there was no significant difference in eGFR, and that at discharge from the hospital, the GFR of patients with sarcopenia already tended to be lower (0.051).

- The following statement on page 13 of the discussion is incorrect “The reduction in muscle mass frequently observed in critical illnesses, like COVID-19, may compromise the reliability of sCr as a primary kidney function marker. However, in our cohort, survivors 284 were assessed 7±2 months after hospital discharge, a period when sCr levels are expected to have reached a steady state, making it a more reliable marker of kidney function than during the acute phase of the critical illness”. At a steady state, if a patient has decreased muscle mass, creatinine will be abnormally low relative to the true GFR, and therefore the creatinine-based estimated GFR will overestimate the GFR.

- It is therefore likely that the prevalence and severity of CKD in the group of patients with sarcopenia is underestimated. This should probably be discussed more explicitly.

- The following two studies could deserve to be cited and briefly discussed in the introduction and/or discussion section:

* Cho et al, Associations of MRI-derived kidney volume, kidney function, body composition and physical performance in ≈38 000 UK Biobank participants: a population-based observational study. Clin Kidney J. 2024 Mar 15;17(4):sfae068. doi: 10.1093/ckj/sfae068.

* Tanaka et al, Trunk muscle quality and quantity are associated with renal volume in nondiabetic people. Clin Kidney J. 2023 Aug 25;16(12):2597-2604. doi: 10.1093/ckj/sfad202. PMID: 38046018; PMCID: PMC10689130.

Reviewers' comments:

Reviewer's Responses to Questions

**Comments to the Author**

1. Is the manuscript technically sound, and do the data support the conclusions?

Reviewer #1: Yes

2. Has the statistical analysis been performed appropriately and rigorously?

Reviewer #1: Yes

3. Have the authors made all data underlying the findings in their manuscript fully available?

Reviewer #1: No

4. Is the manuscript presented in an intelligible fashion and written in standard English?

Reviewer #1: Yes

Reviewer #1: The study by Ribeiro et al. is a single-center cross-sectional study that aimed to assess the relationship between muscle mass and function parameters and kidney health in 743 survivors of the first wave of COVID-19.

While there are published data on muscle-kidney cross-talk, there are no data on this topic in the context of COVID-19, which makes this study original.

My questions and comments are as follows:

- The revised European consensus on the definition and diagnosis of sarcopenia (ref 21) states that ”Although anthropometry is sometimes used to reflect nutritional status in older adults, it is not a good measure of muscle mass [78]. Calf circumference has been shown to predict performance and survival in older people (cut-off point <31 cm) [79]. As such, calf circumference measures may be used as a diagnostic proxy for older adults in settings where no other muscle mass diagnostic methods are available.” There is therefore some uncertainty about the reliability of this method for measuring muscle mass. Since the patients in the study had their quadriceps muscle thickness measured by ultrasound (a method considered reliable for measuring muscle mass), why was the definition of confirmed sarcopenia not based on this ultrasound method?

- The decrease in muscle strength and function may also be due to a change in muscle quality, in addition to muscle quantity (see, for example, 10.1093/ndt/gfy139). Were muscle quality parameters also measured during the quadriceps ultrasound?

- None of the patients in the study had severe sarcopenia. Could this be explained by the following exclusion criterion? “insufficient physical mobility to leave home during follow-up assessments”

- In the results section, on page 10, regarding the following sentence “Table 1 shows that patients with sarcopenia were older (52% vs. 40%; p = 0.002)”, it should be explicitly specified that this refers to patients over the age of 60.

- In the results section, on page 11, regarding the following sentence “Hospital admission and clinical characteristics based on sarcopenia status are described in Supplementary Table S3”, It would be useful to specify in the manuscript that at the time of hospitalization for COVID-19 infection, there was no significant difference in eGFR, and that at discharge from the hospital, the GFR of patients with sarcopenia already tended to be lower (0.051).

- The following statement on page 13 of the discussion is incorrect “The reduction in muscle mass frequently observed in critical illnesses, like COVID-19, may compromise the reliability of sCr as a primary kidney function marker. However, in our cohort, survivors 284 were assessed 7±2 months after hospital discharge, a period when sCr levels are expected to have reached a steady state, making it a more reliable marker of kidney function than during the acute phase of the critical illness”. At a steady state, if a patient has decreased muscle mass, creatinine will be abnormally low relative to the true GFR, and therefore the creatinine-based estimated GFR will overestimate the GFR.

- It is therefore likely that the prevalence and severity of CKD in the group of patients with sarcopenia is underestimated. This should probably be discussed more explicitly.

- The following two studies could deserve to be cited and briefly discussed in the introduction and/or discussion section:

* Cho et al, Associations of MRI-derived kidney volume, kidney function, body composition and physical performance in ≈38 000 UK Biobank participants: a population-based observational study. Clin Kidney J. 2024 Mar 15;17(4):sfae068. doi: 10.1093/ckj/sfae068.

* Tanaka et al, Trunk muscle quality and quantity are associated with renal volume in nondiabetic people. Clin Kidney J. 2023 Aug 25;16(12):2597-2604. doi: 10.1093/ckj/sfad202. PMID: 38046018; PMCID: PMC10689130.

**Do you want your identity to be public for this peer review?** For information about this choice, including consent withdrawal, please see our Privacy Policy

Reviewer #1: **Yes: ** Thomas Stehlé

---

## [Author Response · Author response to Decision Letter 1]

6 Nov 2025

São Paulo, October 2025

To: Dr. Diego Moriconi

Associate Editor

PLOS ONE

Re: PONE-D-25-13841

Dear Editor,

Thank you very much for allowing us to submit a revised version of our manuscript “Skeletal Muscle-Kidney Crosstalk in a Cohort of Critical Illness Survivors”. On behalf of all authors, I would like to thank the editor and the reviewers for the careful analysis of our manuscript and their thoughtful insights. All comments and suggestions have been carefully addressed, and the manuscript was revised and amended accordingly. Please, find below a detailed list of specific responses to each of the points raised by the reviewers. The changes made in the text are highlighted in red so that the revisions can be easily identified.

We are looking forward to hearing from you regarding the status of our revision, which we hope will reach the standards for publication of this esteemed journal. If you have any other questions regarding our manuscript, please, do not hesitate to contact me.

Best regards,

Heitor S. Ribeiro, PhD

Laboratório de Investigação (LIM) 12, Serviço de Nefrologia,

Faculdade de Medicina da Universidade de São Paulo,

São Paulo, Brazil, Postal Code: 01246-903

Office Phone: +551130617343

Email: heitorribeiro@usp.br

---------

Overall changes

“Conflict of interest

EAB received speaker fees from AstraZeneca, Baxter, and Fresenius outside of the submitted work. This does not alter our adherence to PLOS ONE policies on sharing data and materials. Other authors declare that they have no competing interests.”

“Funding

This study receives funding from the Fundação de Amparo à Pesquisa do Estado de São Paulo (FAPESP; grant 22/01769-5) and from Instituto Todos pela Saúde (ITpS; grant C1721). EAB receives a research grant (Bolsa de Produtividade em Pesquisa, 304743/2017-8) from The National Council for Scientific and Technological Development (CNPq). HSR receives a postdoctoral scholarship from FAPESP (24/04564-0). The funders had no role in study design, data collection and analysis, decision to publish, or preparation of the manuscript.”

“Data sharing plan

Data used in this manuscript is part of a large cohort, HCFMUSP COVID-19 Study Group, and is available as Supporting Information.”

“HCFMUSP COVID-19 Study Investigators

Marta Imamura, PhD4; Caroline S. Faria, PhD2; Laura S. Azevedo6; Fábio Augusto R. Gonçalves1. Carlos R. R. Carvalho is the lead author for this group (carlos.carvalho@hc.fm.usp.br).”

-------

Academic Editor

As you will read below, I agree with Reviewer 1 on two major concerns that I believe should be addressed to strengthen the manuscript. First, quadriceps ultrasound, considered a reliable method for assessing muscle mass, was available but not used to define confirmed sarcopenia, whereas calf circumference has known limitations. Second, creatinine-based eGFR likely overestimates kidney function in sarcopenic patients, which may lead to an underestimation of CKD prevalence and severity. This issue should be discussed more explicitly.

Response: Dear academic editor, thank you for the opportunity to improve the quality of our manuscript. We have now incorporated the reviewer’s suggestions.

-------

Reviewer #1

1. Comment: The study by Ribeiro et al. is a single-center cross-sectional study that aimed to assess the relationship between muscle mass and function parameters and kidney health in 743 survivors of the first wave of COVID-19.

While there are published data on muscle-kidney cross-talk, there are no data on this topic in the context of COVID-19, which makes this study original.

1.1 Response: We sincerely thank the reviewer for their time and thoughtful comments on our manuscript. We hope that this revised version meets a higher standard of quality. We believe it is now suitable for publication.

2. Comment: - The revised European consensus on the definition and diagnosis of sarcopenia (ref 21) states that ”Although anthropometry is sometimes used to reflect nutritional status in older adults, it is not a good measure of muscle mass [78]. Calf circumference has been shown to predict performance and survival in older people (cut-off point <31 cm) [79]. As such, calf circumference measures may be used as a diagnostic proxy for older adults in settings where no other muscle mass diagnostic methods are available.” There is therefore some uncertainty about the reliability of this method for measuring muscle mass. Since the patients in the study had their quadriceps muscle thickness measured by ultrasound (a method considered reliable for measuring muscle mass), why was the definition of confirmed sarcopenia not based on this ultrasound method?

2.1 Response: The reviewer is correct about the EWGSOP2 positioning statement regarding anthropometry. In Brazil, but also in most of the countries, there is standardized cut-off values for quadriceps muscle thickness. That was the reason why we adopted the calf circumference as a muscle mass estimate, as the cut-off values adopted by us are widely used and recognized as predictors of sarcopenia in Brazil.

However, we agree with the reviewer that quadriceps muscle thickness measured by ultrasound is a more reliable method. We have now diagnosed sarcopenia based on low quadriceps muscle thickness. By doing so, we had to exclude some cases without quadriceps muscle thickness, and the final analyzed cohort is 734.

2.2 Changes in Methods:

To diagnose low skeletal muscle mass, given the absence of standardized cut-off values for quadriceps muscle thickness in Brazil, we adopted sex-specific lowest tertiles (p33) as follows: (i) rectus femoris <11.6 mm for males and <9.8 mm for females; and (ii) vastus intermedius <11.4 mm for males and <10.3 mm for females.

Sarcopenia diagnosis

The revised European Working Group on Sarcopenia in Older People (EWGSOP2) was adopted to diagnose sarcopenia in the presence of low muscle strength and low skeletal muscle mass25. All patients analyzed had available handgrip strength and ultrasound-measured quadriceps thickness.

2.2 Changes in Results: “Sarcopenia was diagnosed in 21.4% (n = 157) of the survivors.”

3. Comment: - The decrease in muscle strength and function may also be due to a change in muscle quality, in addition to muscle quantity (see, for example, 10.1093/ndt/gfy139). Were muscle quality parameters also measured during the quadriceps ultrasound?- None of the patients in the study had severe sarcopenia. Could this be explained by the following exclusion criterion? “insufficient physical mobility to leave home during follow-up assessments”.

3.1 Response: We did not assess muscle quality parameters. Indeed, this exclusion criterion may explain the absence of severe sarcopenia cases. We have now discussed this as a limitation.

3.2 Changes in Discussion: “The exclusion of survivors with insufficient physical mobility to leave home may have underestimated cases of sarcopenia.”

4. Comment: - In the results section, on page 10, regarding the following sentence “Table 1 shows that patients with sarcopenia were older (52% vs. 40%; p = 0.002)”, it should be explicitly specified that this refers to patients over the age of 60.

4.1 Response: This has been now corrected.

4.2 Changes in the Methods section: “Table 1 shows that patients with sarcopenia were more frequently aged ≥60 years (50% vs. 41%; p = 0.046).”

5. Comment: - In the results section, on page 11, regarding the following sentence “Hospital admission and clinical characteristics based on sarcopenia status are described in Supplementary Table S3”, It would be useful to specify in the manuscript that at the time of hospitalization for COVID-19 infection, there was no significant difference in eGFR, and that at discharge from the hospital, the GFR of patients with sarcopenia already tended to be lower (0.051).

5.1 Response: We have now included this data in Table 1. With the exclusion of some patients for the new analysis based on sarcopenia defined by ultrasound-measured quadriceps thickness, there was no tendency to lower eGFR (p values were 0.235 and 0.252 for admission and discharge, respectively).

4.2 Changes in the Results section: At hospital admission and discharge, eGFR values were similar between groups (p-values = 0.235 and 0.252, respectively).

6. Comments: - The following statement on page 13 of the discussion is incorrect “The reduction in muscle mass frequently observed in critical illnesses, like COVID-19, may compromise the reliability of sCr as a primary kidney function marker. However, in our cohort, survivors 284 were assessed 7±2 months after hospital discharge, a period when sCr levels are expected to have reached a steady state, making it a more reliable marker of kidney function than during the acute phase of the critical illness”. At a steady state, if a patient has decreased muscle mass, creatinine will be abnormally low relative to the true GFR, and therefore the creatinine-based estimated GFR will overestimate the GFR.

- It is therefore likely that the prevalence and severity of CKD in the group of patients with sarcopenia is underestimated. This should probably be discussed more explicitly.

6.1 Response: We agree with the reviewer and have further discussed this.

6.2 Changes in the Discussion section:

“By assuming this bias, it likely overestimates kidney function in survivors with sarcopenia, thus underestimating CKD prevalence and severity.”

7. Comment: - The following two studies could deserve to be cited and briefly discussed in the introduction and/or discussion section:

* Cho et al, Associations of MRI-derived kidney volume, kidney function, body composition and physical performance in ≈38 000 UK Biobank participants: a population-based observational study. Clin Kidney J. 2024 Mar 15;17(4):sfae068. doi: 10.1093/ckj/sfae068.* Tanaka et al, Trunk muscle quality and quantity are associated with renal volume in nondiabetic people. Clin Kidney J. 2023 Aug 25;16(12):2597-2604. doi: 10.1093/ckj/sfad202. PMID: 38046018; PMCID: PMC10689130.

7.1 Response: Although we recognize the importance of these studies for the field, they are not related to our study population. Thank you for the suggestion.

---

## [Decision Letter · Decision Letter 1]

24 Nov 2025

Dear Dr. Ribeiro,

Thank you for submitting your manuscript to PLOS ONE. After careful consideration, we feel that it has merit but does not fully meet PLOS ONE’s publication criteria as it currently stands. Therefore, we invite you to submit a revised version of the manuscript that addresses the points raised during the review process.

We look forward to receiving your revised manuscript.

Kind regards,

Masaki Mogi

Academic Editor

PLOS ONE

Journal Requirements:

Reviewers' comments:

Reviewer's Responses to Questions

**Comments to the Author**

Reviewer #1: All comments have been addressed

2. Is the manuscript technically sound, and do the data support the conclusions?

Reviewer #1: Partly

3. Has the statistical analysis been performed appropriately and rigorously?

Reviewer #1: Yes

4. Have the authors made all data underlying the findings in their manuscript fully available?

Reviewer #1: Yes

5. Is the manuscript presented in an intelligible fashion and written in standard English?

Reviewer #1: Yes

Reviewer #1: I congratulate the authors on the changes made to the manuscript.

The concluding sentence stating that sarcopenia is associated with a lower eGFR (page 15, lines 320 to 322) is not fully supported by the results, since in the binary logistic regression, after adjusting for age, sex, diabetes, and ICU admission, sarcopenia was not associated with eGFR.

I suggest either tempering the conclusion (the conclusion in the abstract is more appropriate)

Or conducting further analyses to try to find and demonstrate a statistically significant link between muscle and eGFR. :

- Multiple linear or polynomial regression to study the impact of sarcopenia, adjusted for other variables, on eGFR.

- Test whether the association between muscle mass (as measured by ultrasound) and eGFR, found in univariate analysis, persists in multivariate analysis.

**Do you want your identity to be public for this peer review?** For information about this choice, including consent withdrawal, please see our Privacy Policy

Reviewer #1: **Yes: ** Dr Thomas Stehlé

---

## [Author Response · Author response to Decision Letter 2]

2 Dec 2025

São Paulo, December 2025

To: Dr. Diego Moriconi

Associate Editor

PLOS ONE

Re: PONE-D-25-13841

Dear Editor,

Thank you very much for allowing us to submit a revised version of our manuscript “Skeletal Muscle-Kidney Crosstalk in a Cohort of Critical Illness Survivors”. On behalf of all authors, I would like to thank the editor and the reviewers for the careful analysis of our manuscript and their thoughtful insights. All comments and suggestions have been carefully addressed, and the manuscript was revised and amended accordingly. Please, find below a detailed list of specific responses to each of the points raised by the reviewers. The changes made in the text are highlighted in red so that the revisions can be easily identified.

We are looking forward to hearing from you regarding the status of our revision, which we hope will reach the standards for publication of this esteemed journal. If you have any other questions regarding our manuscript, please, do not hesitate to contact me.

Best regards,

Heitor S. Ribeiro, PhD

Laboratório de Investigação (LIM) 12, Serviço de Nefrologia,

Faculdade de Medicina da Universidade de São Paulo,

São Paulo, Brazil, Postal Code: 01246-903

Office Phone: +551130617343

Email: heitorribeiro@usp.br

-----

Reviewer #1

1. Comment: I congratulate the authors on the changes made to the manuscript.

1.1 Response: We sincerely thank the reviewer for their time and thoughtful comments on our manuscript. We hope that this revised version meets a higher standard of quality.

2. Comment: The concluding sentence stating that sarcopenia is associated with a lower eGFR (page 15, lines 320 to 322) is not fully supported by the results, since in the binary logistic regression, after adjusting for age, sex, diabetes, and ICU admission, sarcopenia was not associated with eGFR.

I suggest either tempering the conclusion (the conclusion in the abstract is more appropriate)

2.1 Response: We appreciate the reviewer comment and agree with it. The conclusion has been changed.

2.2 Changes in Conclusions:

“In survivors of moderate to severe COVID-19 hospitalization, skeletal muscle mass was associated with eGFR, whereas sarcopenia per se was not independently associated with poor kidney function. Additionally, several muscle mass parameters were significantly associated with eGFR, but not with UACR. These findings suggest an interaction between skeletal muscle and kidney function.”

3. Comment: “Or conducting further analyses to try to find and demonstrate a statistically significant link between muscle and eGFR. :

- Multiple linear or polynomial regression to study the impact of sarcopenia, adjusted for other variables, on eGFR.

- Test whether the association between muscle mass (as measured by ultrasound) and eGFR, found in univariate analysis, persists in multivariate analysis.”

3.1 Response: We have previously conducted such analysis adjusted for confounders. Regarding the multivariate analysis, we believe it is not appropriate because there is high collinearity between skeletal muscle mass markers and physical function, instead we adjusted for clinically relevant confounders such as age, sex, and diabetes.

---

## [Decision Letter · Decision Letter 2]

11 Dec 2025

Skeletal Muscle-Kidney Crosstalk in a Cohort of Critical Illness Survivors

PONE-D-25-13841R2

Dear Dr. Rebeiro,

We’re pleased to inform you that your manuscript has been judged scientifically suitable for publication and will be formally accepted for publication once it meets all outstanding technical requirements.

Kind regards,

Masaki Mogi

Academic Editor

PLOS One

Additional Editor Comments (optional):

Reviewers' comments:

Reviewer's Responses to Questions

**Comments to the Author**

Reviewer #1: All comments have been addressed

2. Is the manuscript technically sound, and do the data support the conclusions?

Reviewer #1: Yes

3. Has the statistical analysis been performed appropriately and rigorously?

Reviewer #1: Yes

4. Have the authors made all data underlying the findings in their manuscript fully available?

Reviewer #1: Yes

5. Is the manuscript presented in an intelligible fashion and written in standard English?

Reviewer #1: Yes

Reviewer #1: (No Response)

**Do you want your identity to be public for this peer review?** For information about this choice, including consent withdrawal, please see our Privacy Policy

Reviewer #1: **Yes: ** Thomas Stehlé

---

## [Editor Report · Acceptance letter]

PONE-D-25-13841R2

PLOS One

Dear Dr. Ribeiro,

I'm pleased to inform you that your manuscript has been deemed suitable for publication in PLOS One. Congratulations! Your manuscript is now being handed over to our production team.

Kind regards,

on behalf of

Dr. Masaki Mogi

Academic Editor

PLOS One